# Improving Continual Learning by Accurate Gradient Reconstructions of the Past

## Abstract

Knowledge reuse is essential for continual learning, and current methods attempt to realize it through regularization or experience replay. These two strategies have complementary strengths, e.g., regularization methods are compact, but replay methods can mimic batch training more accurately. At present, little has been done to find principled ways to combine the two methods and current heuristics can give suboptimal performance. Here, we provide a principled approach to combine and improve them by using a recently proposed principle of adaptation, where the goal is to reconstruct the "gradients of the past", i.e., to mimic batch training by estimating gradients from past data. Using this principle, we design a prior that provably gives better gradient reconstructions by utilizing two types of replay and a quadratic weight-regularizer. This improves performance on standard benchmarks such as Split CIFAR, Split TinyImageNet, and ImageNet-1000. Our work shows that a good combination of replay and regularizer-based methods can be very effective in reducing forgetting, and can sometimes even completely eliminate it.

## 1 Introduction

Continual learning (Parisi et al., 2019) aims for accurate incremental training over a large number of individual tasks/examples. This can potentially reduce the frequency of retraining in deep learning, making algorithms easier to use and deploy, while also reducing their environmental impact (Diethe et al., 2019; Paleyes et al., 2020). The main challenge in continual learning is to remember past knowledge and reuse it to continue to adapt to new data. This can be difficult because the future is unknown and can interfere with past knowledge (Sutton, 1986; Mermillod et al., 2013; Kirkpatrick et al., 2017). Performance, therefore, heavily depends on the strategies used to represent and reuse past knowledge.

Two popular strategies of knowledge reuse are based on regularization and experience replay and have complementary strengths. For example, the well-known Elastic-Weight Consolidation (EWC) (Kirkpatrick et al., 2017), which regularizes the new weight-vector to keep it close to the old one, is compact and requires storing only two vectors, one containing the weights and the other their importance (often the empirical Fisher). A variety of other such regularizers have been proposed (Schwarz et al., 2018; Zenke et al., 2017b; Li & Hoiem, 2018; Nguyen et al., 2018). This is very different from experience replay (Robins, 1995; Shin et al., 2017), where past examples are simply added during future training. Memory cost here can be substantial, but it can boost accuracy if the memory represents the past well. Clearly, combining the two approaches can strike a good balance between performance and memory size.

At present, little has been done to find principled ways to combine the two strategies. Some works have used knowledge distillation (Rebuffi et al., 2017; Buzzega et al., 2020) or functional regularization (Titsias et al., 2020; Pan et al., 2020), where predictions evaluated at the examples in memory are regularized. Such approaches are promising, but it is not clear why the specific choices of regularizers and memory work well and whether there are better choices that lead to further improvements. Here, we aim to fix this issue.

In this paper, we provide a principled approach to combine and improve the two strategies. Our approach is based on a recently proposed principle of adaptation (Khan & Swaroop, 2021), where a prior called Knowledge-adaptation prior (K-prior) is used to reconstruct the gradients of the past

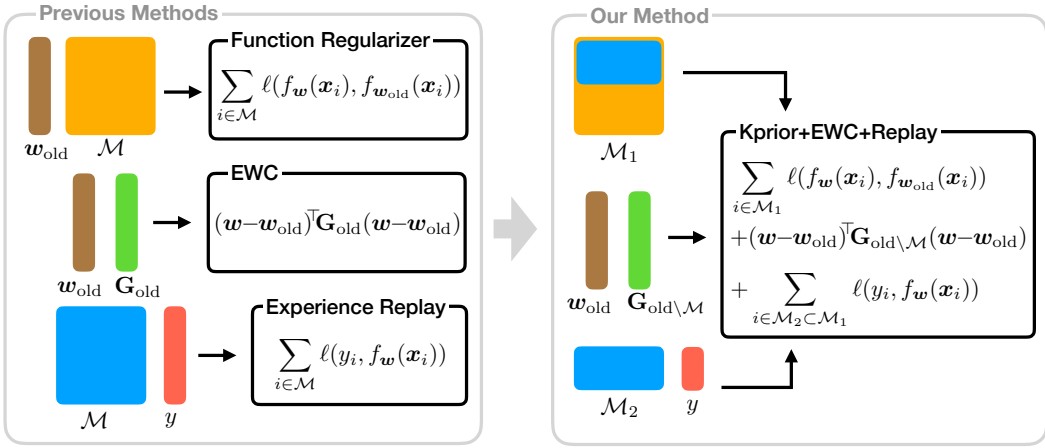

Figure 1: Using the principle of gradient reconstructions, we design a new prior (right) to combine different types of regularization and memory-based methods. EWC uses a quadratic regularizer based on the old weight-vector and its importance, while experience replay uses a memory of past examples along with their labels. Function regularization often does not require the labels, but also lacks a weight regularizer. Our method combines these different types of methods. The notable differences are that our method's importance vector excludes the examples in the memory set, and that experience replay is applied only to the second memory set. The rest of the examples in the memory set do not require labels, and can be compressed to only keep a small set of representative inputs that may not be part of the old training sets.

training objective. In the continual learning setting, this requires an accurate reconstruction of the gradients over all the past tasks, and can be used as a guideline to design better priors. Khan & Swaroop (2021) considered only one-task adaptation, which does not consider the errors accumulated over multiple tasks in problems like continual learning. They also used a simple quadratic regularizer, and did not employ experience replay. Our goal here is to extend their method to multiple tasks and use it to combine different regularization and replay methods.

Using the principle of gradient reconstruction of the past, we design a prior that combines a weight regularizer, a functional regularizer, and experience replay (Fig. 1). Each piece contributes to the reduction of a different type of error. The reduction in the reconstruction error leads to consistent improvements on standard benchmarks for multi-task image classification in task-incremental continual learning setting, such as Split CIFAR, Split TinyImageNet, and ImageNet-1000, across various memory budgets from small to large sizes. The results demonstrate the effectiveness of our approach. The approach is principled and can yield provably better strategies than the current heuristics used in the literature.

## 2 CONTINUAL LEARNING METHODS

We focus on a continual learning (CL) problem to incrementally learn from a sequence of data sets $\mathcal{D}_1, \ldots, \mathcal{D}_T$, corresponding to a total of $T$ tasks. This is different from the commonly used batch-training in deep learning, where data from all the tasks is assumed to be available at all times during training. CL is challenging because the model needs to repeatedly adapt to new tasks, while not forgetting previous-gathered knowledge. Our goal is to get the performance as close as possible to the model that is trained on data from all tasks.

Formally, consider a supervised learning problem with $\mathcal{D}_t$ containing $N$ input-output pairs $(\boldsymbol{x}_i, y_i)$, with $\boldsymbol{x}_i \in \mathbb{R}^D$ and $y_i \in \mathcal{Y}$, and we wish to train a model with output $f_{\boldsymbol{w}}(\boldsymbol{x}_i)$ (also denoted as $f_{\boldsymbol{w}}^i$), and a $P$-length parameter $\boldsymbol{w}$ in a space $\mathcal{W} \subset \mathbb{R}^P$. Then, at any given task $t$, the best possible model parameters $\boldsymbol{w}_t^\star$ can be obtained by training on all the data examples $\mathcal{D}_{1:t} = \cup_{i=1}^t \mathcal{D}_i$, for example,

by solving an optimization problem shown below,

$$\boldsymbol{w}_t^\star = \arg\min_{\boldsymbol{w} \in \mathcal{W}} \ell_t^{\text{batch}}(\boldsymbol{w}), \quad \text{where} \quad \ell_t^{\text{batch}}(\boldsymbol{w}) = \sum_{i \in \mathcal{D}_{1:t}} \ell\big(y_i, h(f_{\boldsymbol{w}}^i)\big) + \mathcal{R}(\boldsymbol{w}). \tag{1}$$

Here, we assume the loss $\ell(y_i, h(f_i))$ to be defined through the log-likelihood of an exponential family distribution (e.g., cross-entropy), with $h(f)$ being a transformation of the model outputs (e.g., softmax) defined using the link-function of the distribution. We denote a regularizer by $\mathcal{R}(\boldsymbol{w})$, which could be either be implicit or explicitly defined, but in what follows, we will use an $L_2$ regularizer $\mathcal{R}(\boldsymbol{w}) = \frac{1}{2}\delta\|\boldsymbol{w}\|^2$, with $\delta \geq 0$.

The main challenge in CL is to remember the useful past knowledge extracted from previous tasks $\mathcal{D}_{1:t-1}$, and reuse it during training over the new task $\mathcal{D}_t$ to get as close as possible to $\boldsymbol{w}_t^\star$. We would like a compact summary of the past knowledge, because we are not allowed to store all past data. We will now briefly describe two strategies used in the literature, based on regularization and experience reply, and discuss the challenges in combining them.

Regularization-based approaches try to keep the new weight-vector close to the old one, with the hope that this will avoid forgetting and facilitate knowledge reuse. For example, the most common is a weight-regularization technique known as Elastic-Weight Consolidation (EWC) (Kirkpatrick et al., 2017), where, at a task $t$, we minimize the following objective using the previous weight $\boldsymbol{w}_{t-1}$,

$$\ell_t^{\text{weight}}(\boldsymbol{w}) = \sum_{i \in \mathcal{D}_t} \ell\big(y_i, h(f_{\boldsymbol{w}}^i)\big) + \frac{\lambda}{2}(\boldsymbol{w} - \boldsymbol{w}_{t-1})^\top \mathbf{F}_{t-1}(\boldsymbol{w} - \boldsymbol{w}_{t-1}), \tag{2}$$

where the second term is a quadratic regularizer with a weight-importance matrix $\mathbf{F}_{t-1}$, and $\lambda \geq 0$ is a trade-off hyperparameter. The simplest choice of $\mathbf{F}_t$ is to use the diagonal of a generalized Gauss-Newton (GGN) matrix (Martens, 2014), where the GGN is defined as,

$$\mathbf{G}(\mathcal{D}_{1:t}) := \sum_{i \in \mathcal{D}_{1:t}} \big[\nabla f_{\boldsymbol{w}_t}^i\big] h'(f_{\boldsymbol{w}_t}^i) \big[\nabla f_{\boldsymbol{w}_t}^i\big]^\top, \tag{3}$$

where $\nabla f_{\boldsymbol{w}_t}^i$ denotes the derivative of $f_{\boldsymbol{w}}(\boldsymbol{x}_i)$ with respect to $\boldsymbol{w}$ at $\boldsymbol{w}_t$ (the Jacobian), and $h'(f^i)$ denotes the derivative of $h(f)$ with respect to $f$ and evaluated at a function value $f^i$. Often, the regularizer $\delta$ is added to the GGN matrix to reduce ill-conditioning. This method uses a compact representation of the past knowledge as we need to only store $\boldsymbol{w}_{t-1}$ and the diagonal of $\mathbf{F}_{t-1}$, which takes $O(P)$ memory size. Computation of $\mathbf{F}_{t-1}$ can be done in an online fashion as the training proceeds over tasks (Schwarz et al., 2018). There exist other choices for the importance vector (Zenke et al., 2017a; Aljundi et al., 2018; Benzing, 2022). This method is also simple to implement within deep-learning codebases, and requires relatively little implementation overhead.

Instead of directly regularizing the weights, experience-replay based methods (Robins, 1995; Shin et al., 2017) simply store a subset of past data in a memory $\mathcal{M}_t$, and add it to the new data during training:

$$\ell_t^{\text{er}}(\boldsymbol{w}) = \sum_{i \in \mathcal{D}_t \cup \mathcal{M}_{t-1}} \ell\big(y_i, h(f_{\boldsymbol{w}}^i)\big).$$

This method can be accurate when the memory represents the data well, but this often requires a large memory that grows with the number of tasks.

A popular approach to combine these two previous approaches is to use functional regularization (Titsias et al., 2020; Pan et al., 2020) where, instead of regularizing the weights, we regularize the function outputs at a few past input locations stored in a memory:

$$\ell_t^{\text{func}}(\boldsymbol{w}) = \sum_{i \in \mathcal{D}_t} \ell\big(y_i, h(f_{\boldsymbol{w}}^i)\big) + \frac{\lambda}{2} \sum_{j \in \mathcal{M}_{t-1}} \ell\left(h(f_{\boldsymbol{w}_{t-1}}^j), h(f_{\boldsymbol{w}}^j)\right). \tag{4}$$

Some methods implement this via knowledge-distillation (Rebuffi et al., 2017; Buzzega et al., 2020), and some also simply use the squared loss instead of using the original loss (Benjamin et al., 2019). An advantage of functional regularization is that it does not require the labels associated with the inputs in $\mathcal{M}_t$, which enables the use of an arbitrary input $\boldsymbol{x}$ which is not restricted to be from $\mathcal{D}_{1:t}$.

For example, we can use a deep generative model to generate pseudo-inputs (Shin et al., 2017), or learn them as in sparse Gaussian processes (Titsias et al., 2020).

Overall, we see that each method has its own complementary strengths. Weight-regularization is compact, experience replay can be accurate, and functional regularization can use arbitrary inputs in memory sets. Combining these approaches can strike a good balance between performance and memory size, but at present, little has been done to find principled ways to combine them. One could simply add them together, but there are many choices one need to make. For example, how should we choose the importance vector, the memory set, and the specific forms of the regularizers? Our goal in this paper is to provide a principled approach to answer such questions.

## 3    PRINCIPLE OF ADAPTATION: GRADIENT RECONSTRUCTION OF THE PAST

We will use the recently proposed principle of adaptation by Khan & Swaroop (2021) to combine and improve the CL strategies discussed in the previous section. The principle suggests to reconstruct the gradient of the past objective by using a combination of weight and function-space regularizers. Specifically, at task $t$, we consider minimizing (for some $\tau > 0$)

$$\ell_t^{\text{K-prior}}(\boldsymbol{w}) = \sum_{i \in \mathcal{D}_t} \ell\big(y_i, h(f_{\boldsymbol{w}}^i)\big) + \tau\, \mathcal{K}(\boldsymbol{w}; \boldsymbol{w}_{t-1}, \mathcal{M}_{t-1}), \tag{5}$$

with $\mathcal{K}(\boldsymbol{w}; \boldsymbol{w}_t, \mathcal{M}_t)$ being a regularizer that combines a weight-space regularizer using $\boldsymbol{w}_t$ and a function-space regularizer over the memory set $\mathcal{M}_t$. The regularizer is called the Knowledge-adaptation prior (or K-prior), and is designed with the goal to minimize the gradient reconstruction error of the past training objectives. We will use $\tau = 1$ unless noted otherwise.

Specifically, at task $t$, the past training objective is $\ell_{t-1}^{\text{batch}}(\boldsymbol{w})$, and we want to design the prior to minimize the magnitude of the gradient error for all $\boldsymbol{w}$:

$$\boldsymbol{e}_t(\boldsymbol{w}) := \nabla \ell_{t-1}^{\text{batch}}(\boldsymbol{w}) - \nabla \mathcal{K}(\boldsymbol{w}; \boldsymbol{w}_{t-1}, \mathcal{M}_{t-1}).$$

The loss $\ell_{t-1}^{\text{batch}}$ depends on all the past data $\mathcal{D}_{1:t-1}$, and our goal is to reconstruct its gradient by using the weight vector $\boldsymbol{w}_{t-1}$ and a memory $\mathcal{M}_{t-1}$.

Khan & Swaroop (2021) showed that many existing adaptive strategies in machine learning for one-step adaptation tasks can be recovered from this principle. For example, for generalized-linear models $f_{\boldsymbol{w}}^i = \boldsymbol{x}_i^\top \boldsymbol{w}$, the error $\boldsymbol{e}_t(\boldsymbol{w})$ is zero when we use the following K-prior with $\mathcal{M}_{t-1} = \mathcal{D}_{1:t}$:

$$\mathcal{K}(\boldsymbol{w}; \boldsymbol{w}_{t-1}, \mathcal{M}_{t-1}) = \sum_{i \in \mathcal{M}_{t-1}} \ell\Big(h(f_{\boldsymbol{w}_{t-1}}^i), h(f_{\boldsymbol{w}}^i)\Big) + \frac{\delta}{2}(\boldsymbol{w} - \boldsymbol{w}_{t-1})^\top(\boldsymbol{w} - \boldsymbol{w}_{t-1}). \tag{6}$$

This is surprising because the K-prior does not use the labels, just like the functional regularization discussed earlier, yet the gradient can be reconstructed by using the predictions at the inputs locations $\boldsymbol{x}_i \in \mathcal{D}_{1:t-1}$. Many other similar results are shown by Khan & Swaroop (2021) for other models, such as support vector machines, Gaussian processes, variational inference, knowledge distillation, functional-regularization, and the memory-based methods used in continual learning.

There are multiple issues with the work of Khan & Swaroop (2021), which we will fix in this paper. First, they did not design any priors for multi-task setup such as continual learning. Second, the error incurred by their K-prior, of form Eq. (6), is non-zero for neural networks: see Khan & Swaroop (2021, Sec. 4.2). For the continual learning problem, this can be disastrous because errors can accumulate quickly over tasks, deteriorating performance. Third, the weight-regularizer in Eq. (6) ignores the weight importance, which is commonly used in other works (Kirkpatrick et al., 2017; Schwarz et al., 2018; Ritter et al., 2018) and can lead to a suboptimal performance.

In this paper, we will fix these issues and use the principle to combine and improve regularization and memory-based methods. We will start with functional-regularization, and then add a weight regularizer and experience replay to decrease its error. This will give us a prior that provably gives better error than each individual method.

## 4 A NEW IMPROVED K-PRIOR

Using the principle described in the previous section, we will now design a prior that combines a weight regularizer, a functional regularizer, and experience replay (Fig. 1). The functional regularizer is based on knowledge distillation, and combined with an EWC-style quadratic regularizer which uses a specific importance vector to minimize the error in the K-prior of Eq. (6). An additional experience-replay term further reduces the error by storing the labels for a subset of the memory set. We will see that each piece contributes to the reduction of a different type of error, and the combination overall gives lower error than each individual component.

### 4.1 THE ERROR IN THE K-PRIOR EQ. (6) WHEN USING A LIMITED MEMORY

We start by analyzing the error in the K-prior of Eq. (6) when it is defined with a limited memory $\mathcal{M}_{t-1}$, instead of the full data $\mathcal{D}_{1:t-1}$. As shown in Appendix A, the error is given as follows,

$$e_t(\boldsymbol{w}) = \underbrace{\sum_{i \in \mathcal{D}_{1:t-1} \setminus \mathcal{M}_{t-1}} \nabla f_{\boldsymbol{w}}^i \left[ h(f_{\boldsymbol{w}}^i) - h(f_{\boldsymbol{w}_{t-1}}^i) \right]}_{:=\boldsymbol{e}^{\mathrm{mem}}(\boldsymbol{w};\boldsymbol{w}_{t-1},\mathcal{D}_{1:t-1} \setminus \mathcal{M}_{t-1})} + \underbrace{\sum_{i \in \mathcal{D}_{1:t-1}} \nabla f_{\boldsymbol{w}}^i r_{\boldsymbol{w}_{t-1}}^i + \delta \boldsymbol{w}_{t-1}}_{:=\boldsymbol{e}^{\mathrm{NN}}(\boldsymbol{w};\boldsymbol{w}_{t-1},\mathcal{D}_{1:t-1})}, \quad (7)$$

where $r_{\boldsymbol{w}_t}^i = h(f_{\boldsymbol{w}_t}^i) - y_i$ is the residual left after prediction. The first error term $\boldsymbol{e}^{\mathrm{mem}}$ arises due to the use of limited memory $\mathcal{M}_{t-1}$, and can be reduced to zero by increasing the memory size to include all the past input examples. In contrast, the second error term $\boldsymbol{e}^{\mathrm{NN}}$ arises due to the use of neural networks, and reduces only when the network gets better in predicting the past data $\mathcal{D}_{1:t-1}$, that is, when the residuals go to zero. We will now show that $\boldsymbol{e}^{\mathrm{mem}}$ can be reduced by adding an EWC-style weight regularizer, while $\boldsymbol{e}^{\mathrm{NN}}$ can be reduced by adding an experience replay term with a specific memory.

### 4.2 REDUCING $\mathbf{e}^{\mathrm{MEM}}$ USING AN EWC-STYLE REGULARIZER

The error $\boldsymbol{e}^{\mathrm{mem}}$ can be reduced by using a first-order Taylor approximation of $h(f_w^i)$ at $\boldsymbol{w}_{t-1}$,

$$h(f_{\boldsymbol{w}}^i) \approx h(f_{\boldsymbol{w}_{t-1}}^i) + h'(f_{\boldsymbol{w}_{t-1}}^i)(\nabla f_{\boldsymbol{w}_{t-1}}^i)^\top (\boldsymbol{w} - \boldsymbol{w}_{t-1}). \quad (8)$$

Plugging this in the definition of the error, we get

$$\boldsymbol{e}_t^{\mathrm{mem}} \approx \underbrace{\left[ \sum_{i \in \mathcal{D}_{1:t-1} \setminus \mathcal{M}_{t-1}} \left[ \nabla f_{\boldsymbol{w}_{t-1}}^i \right] h'(f_{\boldsymbol{w}_{t-1}}^i) \left[ \nabla f_{\boldsymbol{w}_{t-1}}^i \right]^\top \right]}_{=\mathbf{G}(\mathcal{D}_{1:t-1} \setminus \mathcal{M}_{t-1})} (\boldsymbol{w} - \boldsymbol{w}_{t-1})$$

where we use the definition of the GGN matrix given in Eq. (3). The right hand side is equal to the gradient of the an EWC-style regularizer,

$$\boldsymbol{c}_t^{\mathrm{mem}} = (\boldsymbol{w} - \boldsymbol{w}_{t-1})^\top \mathbf{G}(\mathcal{D}_{1:t-1} \setminus \mathcal{M}_{t-1})(\boldsymbol{w} - \boldsymbol{w}_{t-1}), \quad (9)$$

which uses the GGN over the past data but excludes the memory $\mathcal{M}_{t-1}$ from it. We can now define a new K-prior by adding the correction term as follows $\mathcal{K}_t + \boldsymbol{c}_t^{\mathrm{mem}}$, which gives,

$$\mathcal{K}_{\mathrm{cor}}^{\mathrm{mem}}(\boldsymbol{w};\boldsymbol{w}_{t-1},\mathcal{M}_{t-1}) := \sum_{i \in \mathcal{M}_{t-1}} \ell\left( h(f_{\boldsymbol{w}_{t-1}}^i), h(f_{\boldsymbol{w}}^i) \right) + \tfrac{1}{2}(\boldsymbol{w} - \boldsymbol{w}_{t-1})^\top \mathbf{F}_{t-1}(\boldsymbol{w} - \boldsymbol{w}_{t-1}).$$

$$(10)$$

where we define $\mathbf{F}_{t-1} = \mathbf{G}(\mathcal{D}_{1:t-1} \setminus \mathcal{M}_{t-1}) + \delta \mathbf{I}$. This K-prior provably reduces the error introduced in the K-prior of Eq. (6) due to a limited memory. Similarly to Online EWC, we can use a diagonal approximation to the GGN, and update it online.

The new K-prior not only reduces the gradient error, but also is more general because other regularizers are obtained as special cases by changing the memory. For an empty memory $\mathcal{M}_{t-1} = \emptyset$, it reduces to the EWC regularizer of Eq. (2) because then the first term in Eq. (10) disappears and the importance $\mathbf{F}_t$ is the GGN defined over all the past data, plus $\delta \mathbf{I}$. On the other hand, when the memory includes all past data, it reduces to the original K-prior in Eq. (6). When using limited memory, the new K-prior combines the functional and weight regularizer in a way to reduce the error in both EWC and the original K-prior.

### 4.3 REDUCING $e^{\text{NN}}$ USING EXPERIENCE REPLAY

The $e^{\text{NN}}$ term depends on the mistakes made on the past data, and can be corrected by including an additional memory $\mathcal{M}_{2,t-1}$ of the past data where mistakes are significant. We first note that the error is equivalent to the gradient of the following,

$$c_t^{\text{NN}} := \sum_{i \in \mathcal{M}_{2,t-1}} f_{\boldsymbol{w}}^i r_{\boldsymbol{w}_{t-1}}^i + \tfrac{1}{2}\delta \boldsymbol{w}^\top \boldsymbol{w}_{t-1}, \tag{11}$$

when $\mathcal{M}_{2,t-1}$ is set to all the past data. Therefore, by adding $\mathcal{K}_{\text{cor}}^{\text{NN}} = \mathcal{K}_t + c_t^{\text{NN}}$, we reduce the error simply to $\sum_{i \in \mathcal{D}_{1:t-1}\setminus\mathcal{M}_2}\left[\nabla f_{\boldsymbol{w}}^i r_{\boldsymbol{w}_{t-1}}^i\right]$.

### 4.4 K-PRIOR WITH EWC-STYLE REGULARIZER AND EXPERIENCE REPLAY

Our new improved K-prior is obtained by simply correcting the K-prior from Eq. (6) by adding the correction terms, that is, $\mathcal{K}_t^{\text{new}} = \mathcal{K}_t + c_t^{\text{mem}} + c_t^{\text{NN}}$. We make a specific choice of the memory: we choose $\mathcal{M}_{2,t-1}$ to be a subset of $\mathcal{M}_1$. This, as we show now, simplifies the computation and brings experience replay into our new K-prior.

Consider the following two terms taken from $\mathcal{K}_{\text{cor}}^{\text{mem}}$ from Eq. (10) and $\mathcal{K}_{\text{cor}}^{\text{NN}}$ from Eq. (11).

$$\sum_{i \in \mathcal{M}_{t-1}} \ell\left(h(f_{\boldsymbol{w}_{t-1}}^i), h(f_{\boldsymbol{w}}^i)\right) + \sum_{i \in \mathcal{M}_{2,t-1}} f_{\boldsymbol{w}}^i r_{\boldsymbol{w}_{t-1}}^i$$

The gradient of these two can be simplified, when the second memory is a subset of the first one,

$$\sum_{i \in \mathcal{M}_{t-1}} \nabla f_{\boldsymbol{w}}^i \left[h(f_{\boldsymbol{w}}^i) - h(f_{\boldsymbol{w}_{t-1}}^i)\right] + \sum_{i \in \mathcal{M}_{2,t-1}} \nabla f_{\boldsymbol{w}}^i r_{\boldsymbol{w}_{t-1}}^i$$

$$= \sum_{i \in \mathcal{M}_{t-1}\setminus\mathcal{M}_{2,t-1}} \nabla f_{\boldsymbol{w}}^i \left[h(f_{\boldsymbol{w}}^i) - h(f_{\boldsymbol{w}_{t-1}}^i)\right] + \sum_{i \in \mathcal{M}_{2,t-1}} \nabla f_{\boldsymbol{w}}^i \left[h(f_{\boldsymbol{w}}^i) - h(f_{\boldsymbol{w}_{t-1}}^i)\right] + \nabla f_{\boldsymbol{w}}^i r_{\boldsymbol{w}_{t-1}}^i$$

$$= \sum_{i \in \mathcal{M}_{t-1}\setminus\mathcal{M}_{2,t-1}} \nabla f_{\boldsymbol{w}}^i \left[h(f_{\boldsymbol{w}}^i) - h(f_{\boldsymbol{w}_{t-1}}^i)\right] + \sum_{i \in \mathcal{M}_{2,t-1}} \nabla f_{\boldsymbol{w}}^i \left[h(f_{\boldsymbol{w}}^i) - y_i)\right]$$

where in the last line we used the definition of the residual to simplify. This gradient is equal to the gradient of a sum of a functional-regularizer term and an experience replay term,

$$\sum_{i \in \mathcal{M}_{t-1}\setminus\mathcal{M}_{2,t-1}} \ell\left(h(f_{\boldsymbol{w}_{t-1}}^i), h(f_{\boldsymbol{w}}^i)\right) + \sum_{i \in \mathcal{M}_{2,t-1}} \ell\big(y_i, h(f_{\boldsymbol{w}}^i)\big) \tag{12}$$

Therefore, we can rewrite the new K-prior as the following combination,

$$\mathcal{K}^{\text{new}}(\boldsymbol{w}; \boldsymbol{w}_{t-1}, \mathcal{M}_{t-1}) = \underbrace{\sum_{i \in \mathcal{M}_{1,t-1}} \ell\left(h(f_{\boldsymbol{w}_{t-1}}^i), h(f_{\boldsymbol{w}}^i)\right)}_{\text{Functional-regularization}} + \underbrace{\sum_{i \in \mathcal{M}_{2,t-1}} \ell\big(y_i, h(f_{\boldsymbol{w}}^i)\big)}_{\text{Experience replay}}$$

$$+ \underbrace{\tfrac{1}{2}(\boldsymbol{w} - \boldsymbol{w}_{t-1})^\top \mathbf{F}_{t-1}(\boldsymbol{w} - \boldsymbol{w}_{t-1}) + \tfrac{1}{2}\delta \boldsymbol{w}^\top \boldsymbol{w}_{t-1}}_{\text{Weight-regularization}}, \tag{13}$$

where $\mathcal{M}_{1,t-1}$ denotes the $\mathcal{M}_{t-1}$ without the $\mathcal{M}_{2,t-1}$. The new prior combines a weight regularizer, a functional regularizer, and experience replay. The functional regularizer is based on knowledge distillation, and combined with an EWC-style quadratic regularizer which uses an importance vector obtained over all the past data but excluding the memory $\mathcal{M}_{t-1}$. Using this combination, the new prior gives provably lower gradient reconstruction error than each method alone.

In our experiments, we select the memories as follows. We first randomly (uniformly) sample $\mathcal{M}_1$ from all data $\mathcal{D}_{1:t-1}$, and then sample $\mathcal{M}_2$ as a subset of $\mathcal{M}_1$. We generally set $|\mathcal{M}_2| = \tfrac{1}{2}|\mathcal{M}_1|$ for simplicity. We found this to be superior to using all soft labels (i.e. where $\mathcal{M}_1 \cap \mathcal{M}_2 = \emptyset$, as in vanilla K-priors) or all hard labels (i.e. where $\mathcal{M}_1 = \mathcal{M}_2$, as in Replay (Robins, 1995), which is known to perform poorly (Khan & Swaroop, 2021)). We found this simple selection strategy to work surprisingly well, and leave the study of more sophisticated approaches (as well as of settings where $|\mathcal{M}_2| \neq \tfrac{1}{2}|\mathcal{M}_1|$) for future work.

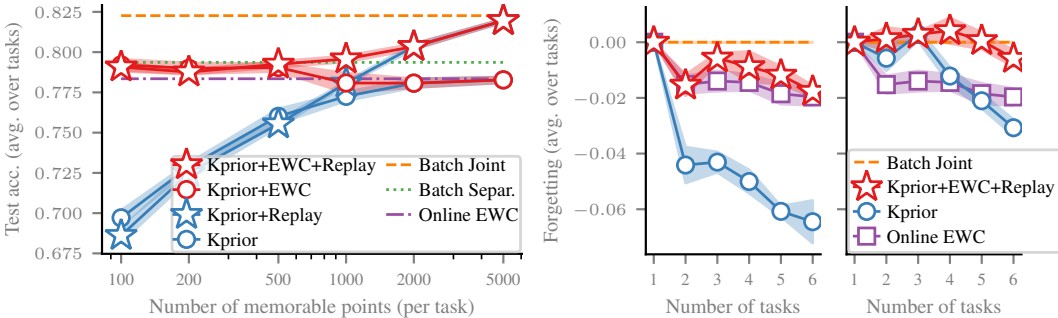

Figure 2: **Results on Split-CIFAR.** Kprior+EWC+Replay is superior across memory sizes, closely approaching Batch Joint for memory size 5,000 (left; x-axis log-scaled). It also forgets less (relative to Batch Joint) with a growing number of tasks, for memory sizes 100 (middle) and 5,000 (right).

## 5 EMPIRICAL EVALUATION

We first describe the general experimental setup used throughout our evaluation. We then present empirical results corroborating the practical efficacy of our proposed Kprior+EWC+Replay method. We focus on multi-class classification in the task-incremental learning setting. In particular, we evaluate on three continual learning benchmarks with increasing size and thus difficulty: 1) Split-CIFAR (medium-scale), 2) Split-TinyImageNet (medium-to-large-scale), and 3) ImageNet-1000 (large-scale). See Appendix B for more details on the experiments (e.g. hyperparameters used).

### 5.1 EXPERIMENTAL SETUP

**CL setup.** We mostly follow previous works on CL. We consider the common multi-head task-incremental (Van de Ven & Tolias, 2019) setting with known task identities: each method is trained sequentially on all tasks with a separate classification head per task, and is told which task an input belongs to both at train and test time. We report test accuracy of the final model trained on the entire task sequence (averaged over the test sets of all observed tasks). We also compute average forgetting (aka backward-transfer) as defined in Lopez-Paz & Ranzato (2017), which captures the (average) difference in accuracy between when a task is first trained and after the final task. We plot mean $\pm$ standard error over three seeds, and assess performance across a wide range of memory sizes.

**Methods.** In addition to our proposed Kprior+EWC+Replay method, we evaluate four relevant baselines for comparison. As our method combines Kprior with both EWC and Replay, we also do ablations to assess the benefit of either term alone. In summary, we consider the following methods:

1. **Kprior+EWC+Replay**. Our proposed regularizer which combines the K-prior function regularizer over $\mathcal{M}_1$ with the EWC-style weight regularizer and the Replay term over $\mathcal{M}_2$.

2. `Kprior`. The original K-prior regularizer over $\mathcal{M}_1$ as proposed by Khan & Swaroop (2021), *without* the EWC-style weight regularization term and *without* the Replay term over $\mathcal{M}_2$.

3. `Kprior+EWC` (ablation). A regularizer combining the K-prior function regularizer over $\mathcal{M}_1$ with *only* the EWC-style weight regularization, i.e. *without* the Replay term over $\mathcal{M}_2$.

4. `Kprior+Replay` (ablation). A regularizer combining the K-prior function regularizer over $\mathcal{M}_1$ with *only* the Replay term over $\mathcal{M}_2$, i.e. *without* the EWC-style weight regularization.

5. `Batch Joint`. Joint batch training of a single multi-head model across the data of all tasks, i.e. the optimal CL solution which serves as an upper bound we wish to approach.

6. `Batch Separate`. Independent batch training of a *separate* model for each task.

7. `Online EWC` (Schwarz et al., 2018), which has the same weight regularizer as Kprior+EWC+Replay, but *without* the function regularizer over $\mathcal{M}_1$ or the Replay term over $\mathcal{M}_2$.

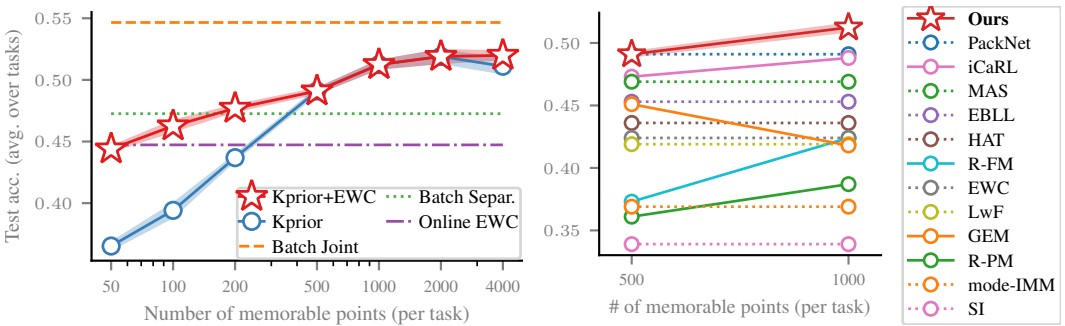

Figure 3: **Results on Split-TinyImageNet.** Kprior+EWC performs well across memory sizes (left; x-axis log-scaled) and compared to further strong baselines from Delange et al. (2021) (right).

## 5.2 RESULTS ON SPLIT-CIFAR

**Setup.** Split-CIFAR (Zenke et al., 2017b) has 6 tasks with 10 classes each. The first task is CIFAR-10 (Krizhevsky et al., 2009) with 50,000 training and 10,000 test data points across 10 classes. The subsequent 5 tasks are taken sequentially from CIFAR-100 (Krizhevsky et al., 2009), each with 5,000 training and 1,000 test data points across 10 classes. In total, we thus have 90,000 data points. We use the same CifarNet model as Zenke et al. (2017b); Pan et al. (2020): a multi-head CNN with 4 convolutional layers, followed by 2 dense layers with dropout, with $\sim$1.2M model parameters in total. On each task, we train for 80 epochs using Adam with learning rate $10^{-3}$ and batch size 256.

**Results.** Fig. 2 shows our results on Split-CIFAR. We consider memory sizes between 100 and 5,000 per task; at 5,000, we thus store 10% of the data for task 1, and all data for tasks 2-5. We see that Kprior performs poorly at small memory sizes and is much worse than Online EWC. While performance improves noticeably with growing memory size, Kprior remains far below Batch Joint even at memory size 5,000. This confirms that when using a NN instead of a GLM, the theory behind Kprior (see Section 3) indeed ceases to hold. However, when we add the Replay term to correct for that error (Kprior+Replay), performance is substantially boosted at large memory sizes, actually enabling us to reach Batch Joint performance for memory size 5,000. However, the Replay correction term does not help at small memory sizes. In contrast, adding the EWC-style weight regularizer (Kprior+EWC) substantially improves performance at small memory sizes, empirically confirming the property that Kprior+EWC converges to Online EWC for small memories (see Section 4.2). However, we also confirm that for large memories, Kprior+EWC converges to Kprior, resulting in a performance drop. Finally, we see that Kprior+EWC+Replay combines the complementary benefits of both error correction terms, i.e. of both EWC-style weight-regularization and Replay, to significantly improve upon vanilla Kprior in both the small *and* large memory regime. Fig. 2 (mid & right) further shows that our method can leverage prior knowledge more effectively than other methods, thereby suffering less from forgetting with a growing number of tasks. This confirms that the two error correction terms in Kprior+EWC+Replay are particularly important for mitigating error accumulation across longer task sequences.

## 5.3 RESULTS ON SPLIT-TINYIMAGENET

**Setup.** Following Delange et al. (2021), we construct Split-TinyImageNet by dividing TinyImageNet (Le & Yang, 2015) into a sequence of 10 tasks with 20 classes each (using the same random division as in Delange et al. (2021)). Each class has 500 data points split into training ($80\%$) and validation ($20\%$), and 50 test points (totalling to 110,000 points). We use the VGG-like BASE model from Delange et al. (2021) with 6 convolutional layers, 4 max pooling layers, and 2 dense layers, with a total of $\sim$3.5M parameters. On each task, we train for 70 epochs (with early stopping and exponential learning rate decay, without regularization) using SGD with momentum 0.9 and batch size 200. This replicates the setup of Delange et al. (2021) to make our results directly comparable.[1]

---

[1]The only difference lies in the hyperparameter tuning procedure: while Delange et al. (2021) use their proposed online tuning algorithm, we resort to a standard grid search for simplicity; see Appendix B for details.

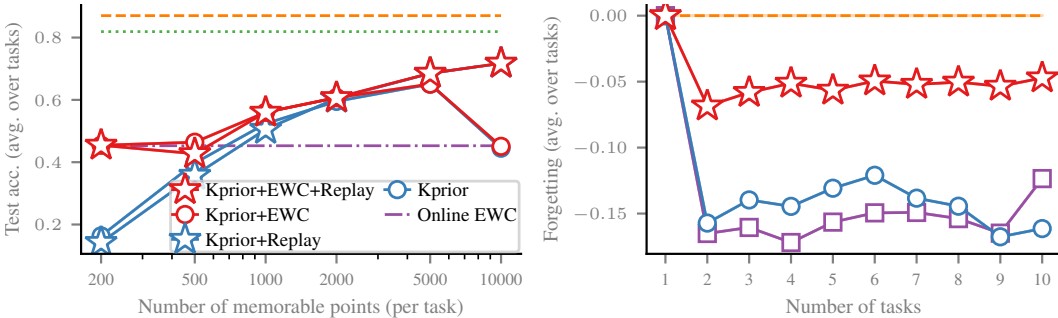

Figure 4: **Results on ImageNet-1000.** Kprior+EWC+Replay performs favorably across a range of memory sizes (left; x-axis log-scaled). It also suffers less from forgetting (relative to Batch Joint) with an increasing number of tasks, here exemplary shown at the largest memory size of 10K (right).

**Results.** Fig. 3 shows our results on Split-TinyImageNet. We found that the Replay error correction term does not help on this benchmark, so we representatively plot just `Kprior` and `Kprior+EWC`.[2] We again see that `Kprior+EWC` can substantially improve over `Kprior` (especially at small memory sizes) and `Online EWC` (Fig. 3 left). It also compares favourably against a diverse range of other strong CL methods across all three CL paradigms: 1) memory/rehearsal – iCaRL (Rebuffi et al., 2017), GEM (Lopez-Paz & Ranzato, 2017), R-FM & R-PM (Delange et al., 2021), 2) weight-regularization – LwF (Li & Hoiem, 2017), EBLL (Rannen et al., 2017), EWC (Kirkpatrick et al., 2017), SI (Zenke et al., 2017a), MAS (Aljundi et al., 2018), mode-IMM (Lee et al., 2017), and 3) architectural – PackNet (Mallya & Lazebnik, 2018), HAT (Serra et al., 2018) (Fig. 3 right).[3]

### 5.4 RESULTS ON IMAGENET-1000

**Setup.** We consider the ImageNet-1000 benchmark proposed by Rebuffi et al. (2017), which randomly (uniformly) splits the full ImageNet dataset Deng et al. (2009) of ~1.2M data points into a sequence of 10 tasks with 100 classes and ~120K data points each. Following Rebuffi et al. (2017), we use a ResNet-18 with ~11M model parameters. For training on each task, we use the ImageNet reference training pipeline (with 40 epoch configuration) of the FFCV library (Leclerc et al., 2022).[4]

**Results.** Fig. 4 shows our results on ImageNet-1000. We consider memory sizes between 200 and 10K per task, where the latter amounts to just under 10% of the data. The observed trends qualitatively match those from previous experiments. In particular, `Kprior` underperforms for small memory sizes, and while it improves with increasing memory, it peaks at a 5K memory and then even starts declining. We hypothesize that this is again due to accumulation of the NN error, which might become more severe with a larger memory as more data points can contribute to the error. This is evidenced by the fact that correcting for the NN error (`Kprior+Replay`) substantially boosts performance at a 10K memory (but it remains poor at small memories). In contrast, `Kprior+EWC` again improves accuracy only for small memories. Finally, `Kprior+EWC+Replay` combines the benefit of both error correction terms to perform well across all memory sizes. It also again forgets less along the task sequence, demonstrating that it better mitigates error accumulation.

## 6 CONCLUSION

We proposed to address the CL problem in a theoretically-grounded way by explicitly approximating the optimal model obtained via batch-training on all tasks jointly. To this end, we developed Kprior+EWC+Replay, which efficiently re-uses prior knowledge by combining principles from function-regularization, weight-regularization, and experience replay. Empirically, we demonstrated the effectiveness and scalability of our method across memory sizes, compared to various baselines.

---

[2]This is likely because almost perfect train accuracy is attained on all tasks (see e.g. Table 14 in Delange et al. (2021)). Thus, $e^{NN}$ in Eq. (7) is close to zero, such that NN error correction cannot boost performance.

[3]Results are from Delange et al. (2021); their total memory sizes [4500, 9000] equal [500, 1000] per task.

[4]For all details of the training procedure, see https://github.com/libffcv/ffcv-imagenet/.

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

## A    DERIVATION OF ERROR FOR K-PRIORS WITH LIMITED MEMORY EQ. (7)

Recall that $f_{\boldsymbol{w}}^i = f_{\boldsymbol{w}}(\boldsymbol{x}_i)$ is a shorthand for the model outputs. For ease-of-notation, we will also use the shorthand $h_{\boldsymbol{w}}^i := h(f_{\boldsymbol{w}}^i) = h(f_{\boldsymbol{w}}(\boldsymbol{x}_i))$ for the model predictions. Consider the following expression for the gradient of the (exponential-family) loss (Khan & Swaroop, 2021),

$$\nabla \ell(y_i, h(f_{\boldsymbol{w}}^i)) = \nabla f_{\boldsymbol{w}}^i \left[ h(f_{\boldsymbol{w}}^i) - y_i \right]. \tag{14}$$

We then have,

$$\nabla \ell_t^{\text{batch}}(\boldsymbol{w}) - \nabla \ell_t^{\text{K-prior}}(\boldsymbol{w})$$

$$= \nabla \left( \ell_t^{\text{batch}}(\boldsymbol{w}) - \ell_t^{\text{K-prior}}(\boldsymbol{w}) \right)$$

$$\overset{(1),(5)}{=} \nabla \left( \sum_{i \in \mathcal{D}_t} \ell\left(y_i, h_{\boldsymbol{w}}^i\right) + \ell_{t-1}^{\text{batch}}(\boldsymbol{w}) - \sum_{i \in \mathcal{D}_t} \ell\left(y_i, h_{\boldsymbol{w}}^i\right) - \mathcal{K}(\boldsymbol{w}; \boldsymbol{w}_{t-1}, \mathcal{M}) \right)$$

$$= \nabla \left( \ell_{t-1}^{\text{batch}}(\boldsymbol{w}) - \mathcal{K}(\boldsymbol{w}; \boldsymbol{w}_{t-1}, \mathcal{M}) \right)$$

$$\overset{(1),(6)}{=} \nabla \left( \sum_{i \in \mathcal{D}_{1:t-1}} \ell\left(y_i, h_{\boldsymbol{w}}^i\right) - \sum_{i \in \mathcal{M}} \ell\left(h_{\boldsymbol{w}_{t-1}}^i, h_{\boldsymbol{w}}^i\right) + \tfrac{1}{2}\delta \|\boldsymbol{w}_{t-1}\|^2 \right)$$

$$= \sum_{i \in \mathcal{D}_{1:t-1}} \nabla \ell\left(y_i, h_{\boldsymbol{w}}^i\right) - \sum_{i \in \mathcal{M}} \nabla \ell\left(h_{\boldsymbol{w}_{t-1}}^i, h_{\boldsymbol{w}}^i\right) + \delta \boldsymbol{w}_{t-1}$$

$$\overset{(14)}{=} \sum_{i \in \mathcal{D}_{1:t-1}} \nabla f_{\boldsymbol{w}}^i \left[ h_{\boldsymbol{w}}^i - y_i \right] - \sum_{i \in \mathcal{M}} \nabla f_{\boldsymbol{w}}^i \left[ h_{\boldsymbol{w}}^i - h_{\boldsymbol{w}_{t-1}}^i \right] + \delta \boldsymbol{w}_{t-1}$$

$$= \sum_{i \in \mathcal{D}_{1:t-1}} \nabla f_{\boldsymbol{w}}^i \left[ h_{\boldsymbol{w}}^i - y_i + h_{\boldsymbol{w}_{t-1}}^i - h_{\boldsymbol{w}}^i + h_{\boldsymbol{w}}^i - h_{\boldsymbol{w}_{t-1}}^i \right] - \sum_{i \in \mathcal{M}} \nabla f_{\boldsymbol{w}}^i \left[ h_{\boldsymbol{w}}^i - h_{\boldsymbol{w}_{t-1}}^i \right] + \delta \boldsymbol{w}_{t-1}$$

$$= \sum_{i \in \mathcal{D}_{1:t-1}} \nabla f_{\boldsymbol{w}}^i \left[ h_{\boldsymbol{w}_{t-1}}^i - y_i \right] + \sum_{i \in \mathcal{D}_{1:t-1}} \nabla f_{\boldsymbol{w}}^i \left[ h_{\boldsymbol{w}}^i - h_{\boldsymbol{w}_{t-1}}^i \right] - \sum_{i \in \mathcal{M}} \nabla f_{\boldsymbol{w}}^i \left[ h_{\boldsymbol{w}}^i - h_{\boldsymbol{w}_{t-1}}^i \right] + \delta \boldsymbol{w}_{t-1}$$

$$= \sum_{i \in \mathcal{D}_{1:t-1} \setminus \mathcal{M}} \nabla f_{\boldsymbol{w}}^i \left[ h_{\boldsymbol{w}}^i - h_{\boldsymbol{w}_{t-1}}^i \right] + \sum_{i \in \mathcal{D}_{1:t-1}} \nabla f_{\boldsymbol{w}}^i \left[ h_{\boldsymbol{w}_{t-1}}^i - y_i \right] + \delta \boldsymbol{w}_{t-1}$$

$$= \sum_{i \in \mathcal{D}_{1:t-1} \setminus \mathcal{M}} \nabla f_{\boldsymbol{w}}^i \left[ h_{\boldsymbol{w}}^i - h_{\boldsymbol{w}_{t-1}}^i \right] + \sum_{i \in \mathcal{D}_{1:t-1}} \nabla f_{\boldsymbol{w}}^i r_{\boldsymbol{w}_{t-1}}^i + \delta \boldsymbol{w}_{t-1}$$

where $r_{\boldsymbol{w}_{t-1}}^i = h_{\boldsymbol{w}_{t-1}}^i - y_i$ is the residual of the $i$'th input using the past model parameters $\boldsymbol{w}_{t-1}$.

## B    EXPERIMENT DETAILS

For all methods in all experiments, we tuned hyperparameters in the common way by conducting a (exponentially-spaced) grid search and evaluating performance on a held-out validation set. In particular, we tune the following hyperparameters: a temperature parameter $T$ to scale the logits in the Replay term (as commonly-done in knowledge distillation, see Khan & Swaroop (2021) for a discussion), a trade-off parameter $\tau$ in front of the K-prior-style function regularization term, and a trade-off parameter $\lambda$ in front of the EWC-style weight regularization term.

We used the following tuned values for those hyperparameters: for Split-CIFAR, we have $T = 2.0$ for all methods, $\tau = 0.1$ for Kprior and Kprior+Replay, $\tau = 0.25$ and $\lambda = 2.0$ for Kprior+EWC and Kprior+EWC+Replay, and $\lambda = 10.0$ for Online EWC. For Split-TinyImageNet, we have $T = 1.0$ and $\tau = 16.0$ for all methods. We found that a fixed $\lambda$ across tasks does not work well for this benchmark, so we tuned a separate $\lambda$ per task, resulting in the sequence $[330, 85, 45, 30, 20, 15, 10, 10]$ for all methods. In contrast to our grid search procedure, Delange et al. (2021) use a dedicated iterative hyperparameter tuning strategy that trades-off plasticity vs. stability. For ImageNet, we used $T = 1.0$, $\lambda = 1.0$ and $\tau = 0.16$ for all methods.

