# OpenReview forum: "Improving Continual Learning by Accurate Gradient Reconstructions of the Past"
_ICLR.cc/2023/Conference — Submitted to ICLR 2023_

### Official Review · Reviewer_jV4b · 2022-10-17

**Confidence:** 4
**Correctness:** 4
**Technical Novelty And Significance:** 3
**Empirical Novelty And Significance:** 2
**Recommendation:** 6

**Clarity, Quality, Novelty And Reproducibility:**

The derivation can be improved, maybe by providing an overview of the reasoning at the beginning (i.e., the paper now requires two readings at least to fully understand the ideas). The practical novelty is limited (see below). The paper is reproducible.

**Strength And Weaknesses:**

I have found the derivation of the three techniques starting from a single principle to be interesting and, as far as I know, novel.

The derivation itself is not always easy to follow. For example, on page 4 they state "For example, for generalized-linear models" and they provide Eq. (6). Later on, they use this equation as the basis for their extensions, which was not clear immediately.

The experiments are comprehensive and show the usefulness of the method, although they do not provide a lot of information about computational time and FLOPS.

My main concern is that the combination of methods is not particularly novel, especially since the proposed derivation does not offer a lot of insights except for a careful choice of the memory itself. In addition, the authors are focusing only on the simple TIL scenario, it is unclear whether this can be extended to more complex (e.g., class-incremental) scenarios.

**Summary Of The Paper:**

The paper focuses on the problem of combining replay models and regularization techniques for task-incremental continual learning (TIL). In the words of the authors, "we design a prior that provably gives better gradient reconstructions by utilizing two types of replay and a quadratic weight-regularizer". The concept of prior is taken from (Khan & Swaroop, 2021), where the focus was on model retraining to, e.g., unlearn data. Intuitively, a prior is a term that is added to the loss (Eq. (5)), whose gradients approximate the past loss function (equation below (5)).

In the TIL case, the correction is the loss computed over a small memory of the past task. The authors show that a series of simplifications bring back a number of known techniques:

1) A functional regularization over the memory (Eq. (6)).
2) An EWC-like term over the past data minus the memory (Eq. (9)).
3) An experience replay term over a subset of the memory (Eq. (12)).

In practice, the algorithm proposed in the paper is a combination of these three (known) terms, with the requirement that the memory for (3) is a subset of the memory for (1), and that (2) is computed over the past data minus the memory itself.

**Summary Of The Review:**

The paper provides an interesting derivation of known ideas from the CL literature, eventually leading to a combination of three known techniques from the literature.

---

### Official Review · Reviewer_ozeD · 2022-10-25

**Confidence:** 5
**Correctness:** 2
**Technical Novelty And Significance:** 2
**Empirical Novelty And Significance:** 2
**Recommendation:** 3

**Clarity, Quality, Novelty And Reproducibility:**

The paper is not easy to follow, and the proposed methods are not well explained. Furthermore, as a result, the proposed methods cannot be a novel approach.

**Strength And Weaknesses:**

**Cons:**

C1. The comment "Weight-regularization is compact, experience replay can be accurate, and functional regularization can use arbitrary inputs in memory sets" is not persuasive and ambiguous to be a strength. Using multiple approaches at once can rather take much more memory and computational costs. It would be better to specify the details on how those approaches can be compatible each other.

C2. The motivations for combining the regularization and experience replay are somewhat weak. Just combining those two approaches are not closely related to solving the fundamental problems in CL. It is not clear what kind of problems the authors try to solve in this paper.

C3. The performance of proposed method is much lower than baselines. Though it uses the memory exemplar, the accuracy is slightly lower than PackNet which does not use any memory exemplars during training.

C4. Is it correct to saying "This K-prior provably reduces the error introduced in the K-prior of Eq. (6) due to a limited memory." which is below (10)? I am not sure adding just Fisher matrix to (6) provable reduces the error

**Summary Of The Paper:**

This paper proposed a method combining two approaches: regularization and experience replay. Using K-prior which tries to reconstruct the gradient information of old tasks, the authors smoothly derive the regularization and experience replay through minimizing the error $e^{mem}$ and $e^{NN}$. In the experiment, the proposed method achieves comparable results to the baselines.

**Summary Of The Review:**

I vote to reject this paper. The motivations are weak, and proposed methods are not be a novel approach. Furthermore, some statements are not justified and not easy to understand.

---

### Official Review · Reviewer_6uVZ · 2022-10-27

**Confidence:** 3
**Correctness:** 3
**Technical Novelty And Significance:** 2
**Empirical Novelty And Significance:** 3
**Recommendation:** 5

**Clarity, Quality, Novelty And Reproducibility:**

The novelty of this paper is not sufficient, because it just combines several existing prior methods. The writing skill of the authors is good, and the paper is easy to follow.

**Strength And Weaknesses:**

Strength

1.The proposed method provides a principled way to combine and improve the regularization and replay methods.

2.The theoretical analysis of the proposed method is detailed and sufficient.


Weaknesses

1.The comparison with other CL methods for task incremental learning is not enough.

2.The experimental design is repetitive and simple, and the amount of information shown by experimental results is not enough.

3.As a combined methods, only one set of components are selected(Kprior、EWC、Replay).


**Summary Of The Paper:**

This paper proposes to address the CL problem by approximating the optimal model obtained via batch-training on all tasks jointly. To achieve this, the Kprior+EWC+Replay is developed to efficiently re-use prior knowledge. Experimental results demonstrate the effectiveness and scalability of the proposed method.

**Summary Of The Review:**

The paper provides a general principle to combine the commonly used CL strategies, but the novelty and the experimental settings can be further improved. I tend to reject it.

---

### Official Review · Reviewer_goJh · 2022-10-30

**Confidence:** 2
**Correctness:** 4
**Technical Novelty And Significance:** 1
**Empirical Novelty And Significance:** 2
**Recommendation:** 3

**Clarity, Quality, Novelty And Reproducibility:**

The paper is well written, and the experiments are easily reproducible from the text.

Novelty is limited as the work combines previous approaches to continual learning.

**Strength And Weaknesses:**

Strenghts:
 - clear presentation of the method
 - the empirical results show improvement over the baselines
 - baselines

Weaknesses:
 - limited novelty
 - small batch of experiments (no RL; baselines were also evaluated in RL setups)

**Summary Of The Paper:**

The paper combines data replay and EWC into a single objective to tackle continual learning of tasks.

Results confirm the benefit of combining an experience replay buffer to replay data from old tasks, and functional priors such as EWC.

Authors perform several ablations on split CIFAR100, split mini imagenet and imagenet 1000.

**Summary Of The Review:**

The work presents an empirical study of combining different methods proposed to deal with catastrophic forgetting in continual learning. Although well presented, the results are not surprising and don't bring new insights about continual learning.

---

### Decision · Program_Chairs · 2023-01-20

**Decision:**

Reject

**Justification For Why Not Higher Score:**

Lack of novelty and clarity and the authors did not reply/address the concerns of the reviewers.

**Justification For Why Not Lower Score:**

N/A

**Metareview: Summary, Strengths And Weaknesses:**

	The paper proposes to address the catastrophic forgetting problem in continual learning by combining two approaches: regularisation and experience replay.
The main concern of the reviewers is the lack of novelty of the approach which combines existing approaches in CL. Additional investigations are needed to provide insight on the approach and highlight its benefits. Moreover, the clarity of the presentation (especially the derivation) could be improved as pointed out by Reviewers jV4b and ozeD.